# Maternal perception of masking in children as a preventive strategy for COVID-19 in Nigeria: A multicentre study

**Ann E. Aronu[1], Josephat M. Chinawa** [1]*, **Obinna C. Nduagubam[2], Edmund N. Ossai[3], Awoere T. Chinawa[4], Wilson C. Igwe[5]**

**1** Department of Paediatrics, College of Medicine, University of Nigeria Enugu Campus, Nsukka, Enugu, Nigeria, **2** Department of Paediatrics, Enugu State University Teaching Hospital, Nsukka, Enugu, Nigeria, **3** Department of Community Medicine, College of Health Sciences, Ebonyi State University Abakaliki, Abakaliki, Nigeria, **4** Consultant Community Physician and Lecturer Enugu State University Teaching Hospital, Nsukka, Enugu, Nigeria, **5** Department of Paediatrics, Nnamdi Azikiwe University Nnewi, Nnewi, Anambra, Nigeria

* josephat.chinawa@unn.edu.ng

## Abstract

### Background

The use of face masks by children for the prevention of COVID 19 is still controversial, especially with regards to who should wear the face mask and at what age.

### Objectives

The study aimed to ascertain the perception of mothers on masking in children as a preventive strategy for COVID-19.

### Methodology

This was a cross-sectional study carried out in two health institutions among 387 mothers who presented with their children for the first time in the hospital during the COVID 19 pandemic. A pretested, semi-structured, interviewer-administered questionnaire which was designed by the researchers was used for data collection.

### Results

Minority (44.7%) of the mothers perceived masking in children as an appropriate measure for the prevention of COVID-19. The frequent reasons given by majority (55.3%) of the mothers for the inappropriateness of face mask in children included perceived difficulty in breathing (38.5%) and the child's readiness to pull it off (29.3%). A significantly higher proportion of the children whose mothers were 35 years and above, 64.2% would *wear face* masks when compared with 31.7% of those whose mothers were < 30 years of age ($\chi^2$ = 28.632, p<0.001). Similarly, a significantly higher proportion (51.0%) of the children who were more than one year of age would wear a face mask when compared with 20.5% of those aged eight days to one year ($\chi^2$ = 19.441, p<0.001). The children whose mothers

**Data Availability Statement:** All relevant data are in the paper and Supporting information files.

**Funding:** The author(s) received no specific funding for this work.

**Competing interests:** The authors have declared that no competing interests exist.

were <30 years were about four times less likely to wear a face mask when compared with those whose mothers were aged 35 years and above. (AOR = 0.273; 95%CI: 0.155–0.478). The children whose fathers have attained tertiary education were about twice less likely to wear face masks when compared with those whose fathers have attained secondary education and less. (AOR = 0.554; 95%CI: 0.334–0.919). Mothers' perception of COVID-19 had no significant influence on children's use of face mask ($\chi^2$ = 2.337, p = 0.127)

## Conclusion

Maternal perception of masking in children as an appropriate strategy for preventing COVID-19 is adjudged low in this study. Right perception is significantly enhanced by maternal educational status, employment and marital status.

## Introduction

COVID -19 pandemic has affected about 215 countries in the world [1]. In Africa, South Africa ranks highest with 650,749 cases and 15,499 deaths while 56,388 cases and 1,083 deaths were recorded in Nigeria. Wearing face mask, social distancing and washing of hands are the only non-pharmaceutical means of prevention [1].

It has been documented that people who were asymptomatic can transmit the novel virus and this can only be reduced by wearing of face mask. Use of face masks are very vital tool for the prevention of COVID 19 even among children, especially when they go to school and get involved in social gatherings, like in churches [2].

Asymptomatic cases can also be seen in children, hence the urgent need for the use of face masks among them. A study among children with COVOD 19 revealed that 15.8% were asymptomatic [3]. There is thus evidence that asymptomatic children have the potential to transmit the infection.

It is pertinent to note that even though the viral load associated with COVID 19 is unknown, it is not known if viral shedding increases with severity [4].

The use of face masks in families and children is imperative and indispensable because of the rapid spread of this novel virus in homes. For instance, in China, it was reported that seven in ten human-to-human transmissions of COVID 19 occurred in homes [5, 6].

The World health organization has noted that household transmission of this virus is the harbinger of community spread [5, 6]. In developing countries where the health service is overwhelmed, people now self-isolate at home and this even increases the risk of spread. A study in China showed that the risk of transmission is high if no measure such as face mask was introduced [6–8]. They suggested home-made masks as being very effective during severe epidemics in preventing infections.

The World Health Organization (WHO) has stated that the use of face masks alone will not protect against COVID-19, but opined that the general public should wear face masks, especially a three-layer- fabric mask; which is essentially a non -medical mask [9, 10].

The World Health Organization also stated that children under five years of age should not wear masks. This stemmed from a "do no harm" point of view, which stated that at five years, children usually achieve significant developmental milestones, especially the fine motor, such as, pincer grasp. This fine motor milestone is needed to appropriately use a mask with minimal assistance [11]. However, some countries may contextualize the use of face masks for children

to 2 or 3 years. Using the "do no harm approach", if the age of two or three years of age is to be used for recommending mask use for children, a well-knitted and coordinated supervision by the mother is needed to ensure the correct use of the mask and to prevent any potential harm associated with mask wearing to the child. Recommendations involving different age groups have been made by WHO with respect to the use of face masks in children, as information on COVID-19 continues to evolve.

Most countries, utilize the global guidance from WHO but contextualize it. For instance, in Nigeria, the Presidential Task Force on COVID-19, Federal Ministry of Health, NCDC and partner organizations have intensified programs on the use of face mask. This includes awareness campaigns in communities to sensitize the people on the need to wear face masks. Various slogans and hashtags such as #MaskOnNaija, #MaskingForAFriend and #TakeRe-sponsibility were used [10, 11].

A knowledge gap exists on this issue under study, as few studies on the use of face masks failed to highlight the impact of age and parental factors on the wearing of face masks in children.

The study aimed to ascertain the perception of mothers on masking in children as a preventive strategy for COVID-19.

## Materials and methods

### Study population

This study was carried out in two health institutions namely the Enugu State University Teaching Hospital, Enugu and Nnamdi Azikiwe University Teaching Hospital Nnewi, Nigeria. Both hospitals are tertiary health institutions and designated as isolation centres in the fight against COVID-19. Mothers of child bearing age who gave consent were recruited in the study while those who refused to give consent were excluded from the study.

### Study area

This study was a prospective and cross-sectional study conducted among mothers who presented with their children to the children's outpatient clinics, paediatric wards, as well as the children emergency room in the period of study.

### Sampling technique

Systematic sampling technique using facility register was used to select clients as they present on each day of data collection. An average of 1186 clients presented in the three Paediatric clinics in the health facility including the Children out-patient department, Children Emergency and Immunization clinics monthly within this COVID-19 period. This number, 1186 served as sampling frame. The sampling interval was determined by dividing the sampling frame (1186) by sample size (384), hence a sampling interval of 3 was used. Every third client was recruited for the study based on the order of registration of clients in each of the clinics, on each day of data collection. The clients who attended the various clinics on the five working days of the week were included in the study.

The information given by the mother was supported by what was observed, as the mothers were present with the children during the study. We did not focus on mother-child dyad of wearing of face mask, though mothers were observed in the study, to be masked. However, the National COVID-19 protocol made it mandatory for adults to wear facemasks in public space, and this was strictly enforced by health facilities and financial institutions. Our observation of children wearing face masks necessitated this study. We did not focus on mother-child dyads

while assessing maternal perception and the practice of wearing face masks. Even though adults were required to wear face masks, there was no such clear rule for the children.

## Sample size estimation

The minimum sample size used in this study was calculated using the formula.

$$N = \frac{Z^2 P (I - P)}{D^2}$$

Where Z = 1.96 i.e. the level of significance

P = proportion of mothers who have knowledge about children wearing face mask (taken as 50%; being a new study)

D = Tolerable error (0.05)

Using the above formula, the minimum sample size is 384

3.5% attrition rate was considered, this brought the final value to 398

## Data collection

A pretested, semi-structured, interviewer-administered questionnaire which was designed by the researchers was used for data collection. The questionnaire was administered to the respondents by trained research assistants. The outcome variables were the proportions of mothers with varying perceptions/knowledge on masking in children as a pandemic antidote. We used the mother's and child's age, marital status, educational level of both mother and father, parity, and occupation of father and mother as independent variables.

## Data analysis

Data entry and analysis were done using IBM Statistical Package for Social Sciences (SPSS) statistical software version 25. Frequency tables and cross-tabulation were generated. Chi- square test of statistical significance and multivariate analysis using binary logistic regression were used in the analysis and the level of statistical significance was determined by a p- value of <0.05. In determining the predictors of children wearing face masks, variables that had a p-value of <0.2 on bivariate analysis were entered into the logistic regression model. The results were reported using adjusted odds ratios (AOR) and 95% confidence interval and the level of statistical significance was determined by a p-value of <0.05. The outcome variable was the wearing of a facemask by children. This was determined by a Yes answer to the question of whether the child wears a face mask. This was confirmed by observation of the wearing of a face mask by the child during the period of data collection. Seven variables were used to assess the perception of COVID-19 among the respondents. Each correct answer by any of the respondents was given a score of one while an incorrect answer attracted a score of zero. Respondents who scored ≥60% of the total score were designated as having a good perception of COVID-19 while those that scored less than 60% of the total score were regarded as having a poor perception of COVID-19.

## Ethical approval

The approval of the Enugu State University of Science and technology Health Research and Ethics Committee was obtained before commencing the study (Reference number: ESUTP/ C-MAC/RA/034/Vol 1 /266).

## Consent

Verbal informed consent was obtained from the respondents in order to participate in this study. Written informed consent was not obtained because some of the participants are not educated. The ethics committees/IRB approved this consent procedure.

## Results

Table 1 shows mothers' perception of masking in children. A minor proportion of the children, 43.5% wore face masks which were all home-made. The major reasons for the children not wearing face masks included perceived difficulty in breathing, 38.2% and child's readiness to pull it off, 29.3%.

Table 2 shows the factors influencing the use of face masks by the children. A significantly higher proportion of the children whose mothers were 35 years and above, 64.2% wear face masks when compared with those whose mothers were <30 years of age, 31.7%. ($\chi^2$ = 28.632, p<0.001). Similarly, a significantly higher proportion of the children who were more than one year of age, 51.0% wear a face mask when compared with those who were aged 8 days to one year, 20.5%. ($\chi^2$ = 19.441, p<0.001). A higher proportion of the children whose mothers who have attained secondary education and below, 48.7% wear a face mask when compared with those whose mothers attained tertiary education, 38.5% and the difference in proportions was found to be statistically significant, ($\chi^2$ = 4.183, p = 0.041). Also, a higher proportion of the children whose fathers attained secondary education and less,50% wear face mask when compared with those whose fathers have attained tertiary education, 35.2% and the difference in proportions was found to be statistically significant, ($\chi^2$ = 8.718, p = 0.003).

Mothers' perception of COVID-19 has no significant influence on children wearing of face mask ($\chi^2$ = 2.337, p = 0.127)

**Table 1. Mothers' perception of masking in children.**

| Variable | Frequency (n = 398) | Percent (%) |
|---|---|---|
| Age of child | (n = 398) | |
| <8 days | 227 | 57.0 |
| 8 days–1year | 73 | 18.3 |
| 1–5 years | 72 | 18.1 |
| 6–12 years | 23 | 5.8 |
| >12 years | 3 | 0.8 |
| **Gender of the child** | | |
| Male | 220 | 55.3 |
| Female | 178 | 44.7 |
| Do your child wear a face mask | | |
| Yes | 173 | 43.5 |
| No | 225 | 56.5 |
| Reason for not wearing mask | (n = 225) | |
| Perceived difficulty in breathing | 86 | 38.2 |
| Pulls it off | 66 | 29.3 |
| Cries when wearing a mask | 43 | 19.1 |
| Sickly feeling | 11 | 4.9 |
| Feels child is too young | 10 | 4.4 |
| Corona not real/don't see the need | 9 | 4.0 |

**Table 2. Factors that influence the use of face masks by the children.**

| Variable | Child wears mask | | $\chi^2$ | p value |
|---|---|---|---|---|
| | (n = 398) | | | |
| | Yes N (%) | No N (%) | | |
| Age of Mother | | | | |
| <30 years | 51 (31.7) | 110 (68.3) | 28.632 | <0.001 |
| 30–34 years | 52 (40.6) | 76 (59.4) | | |
| ≥35 years | 70 (64.2) | 39 (35.8) | | |
| Age of child | | | | |
| <8 days | 108 (47.6) | 119 (52.4) | 19.441 | <0.001 |
| 8 days– 1 year | 15 (20.5) | 58 (79.5) | | |
| >1 year | 50 (51.0) | 48 (49.0) | | |
| Gender of baby | | | | |
| Male | 94 (42.7) | 126 (57.3) | 0.110 | 0.741 |
| Female | 79 (44.4) | 99 (55.6) | | |
| Marital status | | | | |
| Married | 149 (42.3) | 203 (57.7) | 1.604 | 0.205 |
| Single ** | 24 (52.2) | 22 (47.8) | | |
| Educational attainment of Mother | | | | |
| Tertiary education | 79 (38.5) | 126 (61.5) | 4.183 | 0.041 |
| Secondary education and less | 94 (48.7) | 99 (51.3) | | |
| Employment status of Mother | | | | |
| Unemployed | 16 (28.1) | 41 (71.9) | 6.431 | 0.040 |
| Self-employed | 105 (46.3) | 122 (53.7) | | |
| Salaried employment | 52 (45.6) | 62 (54.4) | | |
| Educational attainment of Father | | | | |
| Tertiary education | 62 (35.2) | 114 (64.8) | 8.718 | 0.003 |
| Secondary education and less | 111 (50.0) | 111 (50.0) | | |
| Fathers' employment status | | | | |
| Unemployed | 7 (38.9) | 11 (61.1) | 0.163 | 0.922 |
| Self-employed | 126 (43.8) | 162 (56.3) | | |
| Salaried employment | 40 (43.5) | 52 (56.5) | | |
| Mothers' perception of COVID-19 | | | | |
| Good | 69 (39.2) | 107 (60.8) | 2.337 | 0.127 |
| Poor | 104 (46.8) | 118 (53.2) | | |

**Never married, widowed, separated/divorced

Table 3 shows the predictors of a child's use of a face mask. The children whose mothers were <30 years were about four times less likely to wear face mask when compared with those whose mothers were aged 35 years and above. (AOR = 0.273; 95%CI: 0.155–0.478). Similarly, the children whose mothers were between 30–34 years were 2.3 times less likely to wear a face mask when compared with those whose mothers were 35 years and above. (AOR = 0.436; 95% CI: 0.252–0.755. The children who were aged 8 days to one year were about four times less likely to wear a mask when compared with those who were > 1-year old. (AOR = 0.274; 95% CI: 0.132–0.567). The children whose fathers have attained tertiary education were about twice less likely to wear face masks when compared with those whose fathers have attained secondary education and less. (AOR = 0.554; 95%CI: 0.334–0.919).

**Table 3. Predictors of the use of face mask by the children.**

| Variable | Adjusted odds ratio | Value | 95% Confidence Interval | |
|---|---|---|---|---|
| | | | Lower | Upper |
| Age of Mother | | | | |
| <30 years | 0.273 | <0.001 | 0.155 | 0.478 |
| 30–34 years | 0.436 | 0.003 | 0.252 | 0.755 |
| ≥35 years | 1 | | | |
| Age of child | | | | |
| <8 days | 0.810 | 0.420 | 0.510 | 1.400 |
| 8 days– 1 year | 0.274 | <0.001 | 0.132 | 0.567 |
| >1 year | 1 | | | |
| Educational attainment of Mother | | | | |
| Tertiary education | 0.715 | 0.257 | 0.401 | 1.276 |
| Secondary education and less | 1 | | | |
| Employment status of Mother | | | | |
| Unemployed | 0.737 | 0.443 | 0.338 | 1.608 |
| Self-employed | 0.796 | 0.427 | 0.454 | 1.397 |
| Salaried employment | 1 | | | |
| Educational attainment of Father | | | | |
| Tertiary education | 0.554 | 0.022 | 0.334 | 0.919 |
| Secondary education and less | 1 | | | |
| Mothers' perception of COVID-19 | | | | |
| Good | 0.786 | 0.270 | 0.504 | 1.227 |
| Poor | 1 | | | |

Mothers' perception of COVID-19 has no significant correlation with prediction of children wearing face mask ($\chi^2$ = 0.504, p = 1.227)

## Discussion

This study has shown varying maternal perceptions/knowledge of masking in children. It is very tasking for children who have no COVID to put on face masks, and they are likely to take them off since they make them uncomfortable. Children less than 2 years may not benefit from wearing a face mask but may rather be at risk of suffocation. This is because they have small airways, which can make them struggle when breathing and this can cause asphyxiation and even death [12]. The American Academy of Paediatrics has advised the washing of hands and social distancing for these groups of children [13]. Several studies have noted the efficacy and potency of face mask, especially surgical masks and N95 in warding off the transmission of Corona virus. These masks are noted to have the efficacy of protection against the pandemic at 68% and 91%, respectively [14]. Home-made masks were noted to be worn by children who were on masks during the study. Some studies have ascertained the efficacy of home-made masks [15]. They noted that some household materials, vacuum cleaner filters, and tea towels were very effective in the prevention of COVID 19 transmission but stated that they can cause a pressure drop, which makes breathing difficult. They recommended the use of 100% cotton t-shirts and pillowcases which have a filtration efficiency of 51% and 57%, respectively [15]. Linen was also reported to have an effectiveness of 61% [15]. Furthermore, they noted Quilting fabric as the best household fabric, followed by 600-count pillowcases and flannel. Cotton and bandanas were seen to have good tolerability and wearability and associated increased compliance, but with a decreased efficacy [15]. Esposito et al [3] noted that a home-made mask is important in curbing the possibility of transmission of the infection by asymptomatic carriers.

They opined that homemade or purchased cloth masks are suitable for a child who is above the age of two to wear, but the right fit must be guaranteed.

It is gratifying to note that when a facemask is worn correctly, it alters and slows the propelling force of particles expelled from expectoration and prevents transmission [16, 17].

This study also showed that a good number of children did not wear face masks as a method of prevention of Corona virus infection. They hazarded reasons that these children have difficulty in breathing and readily pulled off the face masks.

The benefits of using face masks in children should be weighed against possible harm associated with it. This includes feasibility and discomfort, as well as the effect on interactions and social communication. Besides, the age of the child, socio-cultural correlates and the presence of adult supervision should be considered when masking a child [18].

There has been a lot of controversy surrounding the pattern and clinical profile of COVID 19. For instance, recently, the World Health Organization stated that the number of asymptomatic carriers was very small and that they could not transmit infection [19]. Recently, data showed that 50–75% of subjects with positive throat swab were totally asymptomatic [20]. It is important to note that asymptomatic cases are commonly seen among the paediatric population.

A study has shown that the use of a face mask is very effective in the prevention of COVID-19 in children. Though the effectiveness of homemade mask is not certain, it is documented that surgical masks prevent the inhalation of large droplets, but may not filter submicron-sized airborne particles [21]. It has been reported that asymptomatic people can transmit the new coronavirus 2019 and become important sources of COVID-19. The Universal use of face masks, hand hygiene and physical distancing is very useful in combating this novel virus. To reduce the role of asymptomatic people in COVID-19, it is expedient to prepare the child to use face masks. It is advisable to thoroughly educate the child on the need for compliance, without actually forcing it on him [3, 22–24].

Though a significant proportion of children who were more than one year of age would wear face masks when compared with those who were aged 8 days to one year, we noted that a good number of children less than two years still wore face masks.

This practice is not in keeping with WHO recommendations for wearing face masks in children. The WHO recommended the "do no harm" phenomenon on children wearing face mask. They stated that there is no need for a mask in children less than 5 years of age [25]. However, some countries may contextualize the use of face masks for children to 2 or 3 years [25]. If the age of two or three years of age is to be used for recommending mask use for children, a well-knitted and coordinated supervision is needed to ensure the correct use of the mask and to prevent any potential harm associated with mask use in the child [3]. It is important to note that for children between six and eleven years of age, a risk-based approach should be applied. This includes the intensity of transmission in the locale and availability of data on the risk of infection and transmission in that age group [3, 26]. Other factors such as beliefs, behaviour, customs or social norms and the child's capacity to comply with the appropriate use of masks and availability of appropriate adult supervision should be considered [3, 24, 25]. For children and adolescents 12 years or older, WHO guidelines for mask use in adults should be used [26].

The misconception of children aged 8 days to 1 year would wear face masks as seen in this study, is an eye-opener that will help alert the government to strengthen the policy on the acceptable age of wearing a face mask as well as organize programs geared towards health education for all especially the mothers.

Besides, the majority of children whose mothers have attained at least secondary education wear face masks when compared with those whose mothers attained tertiary education. This

study also showed that a higher proportion of the children whose fathers attained secondary education and less would wear face masks when compared with those whose fathers have attained tertiary education. Health education and promotion on the need for children to wear a face mask, at least for those above two years of age, should be intensified.

This study showed that mothers above 35 years of age, those who are married, whose husbands are employed and who had high educational status prefer that their children should wear a face mask as a preventive measure against Corona virus infection. Studies in Australia have shown that people aged 16–34 years are more reluctant to wear face masks [27], while 68% of subjects aged 50–59 years perceived wearing face masks as very commendable. In keeping with our study, Tang [28] et al, also noted a higher level of compliance among married people when compared with their unmarried counterparts. Taylor [29] et al, also noted that people who never married had very low compliance levels as regards wearing face mask [3]. Nevertheless, getting married increases with age, it could be that age/duration of the marriage is a better predictor of wearing of face mask. Some studies have opined that higher education was strongly associated with wearing face masks. They noted that participants with at least a senior high school education certificate were found to be more likely to wear face masks [29, 30].

We noted a higher proportion of children whose fathers attained secondary education and less, wear a face mask when compared with those whose fathers have attained tertiary education. A study showed that men, in general, don't wear face mask especially when wearing a face covering is not mandatory [24], but they tend to wear a mask when wearing a face covering is mandatory. In Nigeria, there is no penalty or any punitive measures for not wearing a face mask. These highly educated men and those in tertiary institutions know that there is no law and will not bother to wear face masks and they could extend the same apathy to their children. However, their counterparts with secondary education may be ignorant of the non-punitive measures and will prefer to wear a mask out of fear of breaking the law [31].

This study showed that mothers' perception of COVID-19 has does not influence children's use of face masks. This simply means that mothers' perception or knowledge of COVID 19 did not really affect their practice on their children wearing face masks. This gap in perception of COVID 19 and practice of prevention by use of masks in children could be closed by improving behaviour change through campaigns, to reach everyone with appropriate information and guidance. Communication and information dissemination should be intensified.

## Conclusion

Maternal perception of masking in children as an appropriate strategy for preventing COVID-19 is adjudged low in this study. Right perception is significantly enhanced by maternal educational status, employment status and marital status. Furthermore, there is a misconception with regards to mothers putting on face masks on children less than 1 year. Mothers' perception of COVID-19 had no influence on children's use of face masks.

## Recommendation

Children's use of face masks below the age of two years should be discouraged. There should be a clarion call on the government to intensify efforts geared towards making policies on public health promotion, as well as on non- governmental agencies to provide accurate information on the use of facemasks in children.

## Limitation

We did not ascertain the practice of masking on mother-child pair.

## Supporting information

**S1 File.**
(SAV)

**S2 File.**
(DOCX)

## Acknowledgments

We acknowledge the nurses and the hospital clerk who helped in the interview and questionnaire distribution.

## Author Contributions

**Conceptualization:** Ann E. Aronu, Josephat M. Chinawa.

**Data curation:** Ann E. Aronu, Josephat M. Chinawa, Obinna C. Nduagubam, Edmund N. Ossai, Wilson C. Igwe.

**Formal analysis:** Edmund N. Ossai.

**Methodology:** Ann E. Aronu, Josephat M. Chinawa.

**Resources:** Ann E. Aronu, Obinna C. Nduagubam.

**Supervision:** Ann E. Aronu, Josephat M. Chinawa.

**Visualization:** Awoere T. Chinawa.

**Writing – review & editing:** Awoere T. Chinawa, Wilson C. Igwe.

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
