## [Decision Letter · Decision Letter 0]

19 Aug 2020

PONE-D-20-19085

Maternal factors and predictors of wearing face mask on children, in the prevention of COVID 19. A multicentre study

PLOS ONE

Dear Dr. Josephat Chinawa, 

Thank you for submitting your manuscript to PLOS ONE. After careful consideration, we feel that it has merit but does not fully meet PLOS ONE’s publication criteria as it currently stands. Therefore, we invite you to submit a revised version of the manuscript that addresses the points raised during the review process.

Please submit your revised manuscript by  10 September .  If you will need more time than this to complete your revisions, please reply to this message or contact the journal office at plosone@plos.org. Please include the following items when submitting your revised manuscript:

We look forward to receiving your revised manuscript.

Kind regards,

Francesco Di Gennaro

Academic Editor

PLOS ONE

Additional Editor Comments:

Dear Authors,

Below the reviewer suggestions

a) Did participants provide their written or verbal informed consent to participate in this study?

3. Please include additional information regarding the survey or questionnaire used in the study and ensure that you have provided sufficient details that others could replicate the analyses.

For instance, if you developed a questionnaire as part of this study and it is not under a copyright more restrictive than CC-BY, please include a copy, in both the original language and English, as Supporting Information. 

If the original language is written in non-Latin characters, for example Amharic, Chinese, or Korean, please use a file format that ensures these characters are visible.

4. Please state whether you validated the questionnaire prior to testing on study participants. Please provide details regarding the validation group within the methods section.

7. Please include your tables as part of your main manuscript and remove the individual files.

Please note that supplementary tables should be uploaded as separate "supporting information" files

Reviewers' comments:

Reviewer's Responses to Questions

**Comments to the Author**

1. Is the manuscript technically sound, and do the data support the conclusions?

Reviewer #1: Partly

Reviewer #2: No

2. Has the statistical analysis been performed appropriately and rigorously? 

Reviewer #1: No

Reviewer #2: I Don't Know

3. Have the authors made all data underlying the findings in their manuscript fully available?

Reviewer #1: No

Reviewer #2: No

4. Is the manuscript presented in an intelligible fashion and written in standard English?

Reviewer #1: Yes

Reviewer #2: No

5. Review Comments to the Author

Reviewer #1: Thank you for raising the important issue of children wearing a face mask and the maternal predictive factors. This is an important and very timely issues.

Authors have done a good analysis of literature but what is lacking is the policy guidelines in Nigeria. based on WHO or other recommendations- what has the government mandated and disseminated. this is not described. Most countries utilize the global guidance from WHO but contextualize it.

Authors claim that this study will establish the need to wear a mask. While the authors have quoted from the literature about the need to wear a mask, i don't think the analysis lends itself to convincing people on the need t wear a mask.

The question that authors are seeking to answer should be further clarified. Is it the proportion of women who have the knowledge that face mask is essential for children? You write- proportion of mothers who have knowledge of child wearing of face mask. Please describe in detail how you assess the knowledge and the perception.

In the statistical analysis - did you look for variation in young mothers by educational status and also for those above 35 by educational status? could be highly correlated. The age and education status Do you see the same predictive factors? It would have been interesting to see f the wearing of the masks by the mothers themselves also had the similar pattern.. Interesting to see that tertiary education of the father has the opposite effect on wearing a mask by the child. Could the authors explain why?

There is no study limitations mentioned. Please add. in addition- please strengthen the conclusion and what the authors would like to see change as a result of this study. Please add some recommendations for policy and programs

Reviewer #2: General comments.

The authors of this manuscript essentially conducted a KAP study on mothers regarding the wearing of facemask in their children. It was a timely multi-centre, cross-sectional descriptive study in a sub-Saharan African setting which is not spared by the global COVID-19 pandemic. Thus, establishing the KAP on this important preventive strategy of the pandemic is relevant. However, there are some major and minor concerns about this manuscript.

Major concerns

1. Under the Abstract and title, the authors need to note these flaws: The title requires modification for clarity. It is not reflective of the content of the manuscript. The background did not provide the research gap to be filled. The study aim was not explicit enough (it was like a repetition of the study title). For methodology, pertinent elements like sample size, study instrument and how it was administered were missing. Reportage of the results should be improved. The conclusion is not in sync with the results

2. Introduction- The first sentence which reads- ‘Corona Virus infection (COVID 19) has now attained community infection and is now very common in families’- appears rather ambiguous. I think corona virus infection is not synonymous with COVID-19 (the latter is the disease caused by the former). Do the authors mean that the pandemic is now at the level of community transmission? Although the authors tried to highlight what is already known about the topic and the research question, the aim of the study as stated here (‘ascertaining the perception of mothers on the use of face mask in children in this COVID era’) is different from the aim as stated under the Abstract.

3. Methods- The authors revealed the study instrument here as structured self-administered questionnaires. A concise detail of the questionnaire items would have enlightened the reader. Would all the respondents have understood the items? An interviewer-administered questionnaire could have been a better approach.

4. Results- The ‘practice’ aspect of this study namely ‘the wearing of face mask in children’ was based on proxy-report and not observational. Thus, the information provided by these mothers on the proportion of their children who did wear face mask may not be reliable. This underscores the fact that an observational study of mother-child dyads would have been better. For instance, the authors reported that children aged 8 days to 1 year wore face masks. The practice is even contradictory to the reported WHO recommendation in children. The authors reported some socio-demographic variables as predictors of the practice of wearing face mask in children. Curiously, they were silent on the role of knowledge and perception. Maternal knowledge and perception ought to influence their practice of using face mask on their children

5. Discussion- The content and flow of this section should be improved. The authors did not highlight any limitations of the current study. The conclusion was not precise enough but was too concise to have conveyed the message of the study.

6. PLOS authors have the option to publish the peer review history of their article (what does this mean?). If published, this will include your full peer review and any attached files.

Reviewer #1: No

Reviewer #2: No

---

## [Author Response · Author response to Decision Letter 0]

21 Aug 2020

Department of pediatrics,

 College of Medicine,

 University of Nigeria Enugu Campus (UNEC),

 University of Nigeria Teaching Hospital,

 PMB 01129,

 Postal Code 400001, ENUGU, NIGERIA.

 20th August, 2020

Editor-in-Chief,

Plos One

Dear Sir,

MANUSCRIPT CORRECTION: RESPONSES TO EDITOR’S COMMENTS

The authors are grateful to you, all the members of the Editorial Board and the 

reviewers for giving us the opportunity to make corrections on our manuscript titled”

Maternal factors and predictors of wearing face mask on children, in the prevention of COVID 19. A multicentre study.”

Please find below point-by-point corrections made, based on the editor’s comments: 

Editors comment

Authors Response

We have done this.

Editors comment

a) Did participants provide their written or verbal informed consent to participate in this study?

Authors Response

We have added consent to participate section. See page 8 line 168-171

Editors comment

Authors Response

We have explained why we obtained verbal consent. See page 8 line 172-175

Editors comment

3. Please include additional information regarding the survey or questionnaire used in the study and ensure that you have provided sufficient details that others could replicate the analyses.

Authors Response

We have included additional details with respect to questionnaire used in the study. See page 7 line 148-150 

Editors comment

For instance, if you developed a questionnaire as part of this study and it is not under a copyright more restrictive than CC-BY, please include a copy, in both the original language and English, as Supporting Information. 

Authors Response

We have added a copy of the questionnaire as Supporting Information. 

Editors comment

4. Please state whether you validated the questionnaire prior to testing on study participants. Please provide details regarding the validation group within the methods section.

 Authors Response

A pretested, semi-structured, interviewer-administered questionnaire was used. See page 7 line 148-150 

Editors comment

Editors comment

 7. Please include your tables as part of your main manuscript and 

Authors Response

We have included tables in the manuscripts and we have also removed the individual files. See page 9-13

Editors comment

Please note that supplementary tables should be uploaded as separate "supporting information" files

Authors Response

We have taken note of that

Reviewer’s comments

Reviewer #1: Thank you for raising the important issue of children wearing a face mask and the maternal predictive factors. This is an important and very timely issues.

Authors response

Thanks for the commendation

Reviewer’s comments

Authors have done a good analysis of literature but what is lacking is the policy guidelines in Nigeria. based on WHO or other recommendations- what has the government mandated and disseminated. this is not described. Most countries utilize the global guidance from WHO but contextualize it.

Authors response

We have added the WHO recommendations of wearing face mask in introduction and discussion. See page 5 line 95-102. We have also added the WHO recommendations in the Nigerian context. See page 5 line 97-102 

Reviewer’s comments

Authors claim that this study will establish the need to wear a mask. While the authors have quoted from the literature about the need to wear a mask, i don't think the analysis lends itself to convincing people on the need t wear a mask.

Authors response

This statement has been expunged

Reviewer’s comments

The question that authors are seeking to answer should be further clarified. Is it the proportion of women who have the knowledge that face mask is essential for children? You write- proportion of mothers who have knowledge of child wearing of face mask. Please describe in detail how you assess the knowledge and the perception.

Authors response

The focus is use of facemask by the child. The mothers’ perception of COVID-19 has been included. See table 2 and 3

Reviewer’s comments

In the statistical analysis - did you look for variation in young mothers by educational status and also for those above 35 by educational status? could be highly correlated. The age and education status Do you see the same predictive factors? It would have been interesting to see f the wearing of the masks by the mothers themselves also had the similar pattern.. Interesting to see that tertiary education of the father has the opposite effect on wearing a mask by the child. Could the authors explain why?

There is no study limitations mentioned. Please add. in addition- please strengthen the conclusion and what the authors would like to see change as a result of this study. Please add some recommendations for policy and programs

Authors response

Information on wearing of facemask by mothers was not obtained. (This is now regarded as a limitation of the study). The perception of COVID-19 by the mothers was the focus for the mothers. In determining the factors affecting child use of facemask, the age of the mothers was categorized into three, <30 years, 30-34 years and 35 years and above. The educational attainment of the mothers and fathers of the children was categorized into tertiary education and secondary education and less.

We have explained why tertiary education of the father has the opposite effect on wearing a mask by the child. We have added limitation to the study, broadened the conclusion and we have mentioned recommendation for policies and program. See page 16 line 313-322

Reviewer’s 2 comments

Reviewer #2: General comments.

The authors of this manuscript essentially conducted a KAP study on mothers regarding the wearing of facemask in their children. It was a timely multi-centre, cross-sectional descriptive study in a sub-Saharan African setting which is not spared by the global COVID-19 pandemic. Thus, establishing the KAP on this important preventive strategy of the pandemic is relevant. However, there are some major and minor concerns about this manuscript.

Authors response

Thanks for the commendation

Reviewer’s 2 comments

Major concerns

1. Under the Abstract and title, the authors need to note these flaws: The title requires modification for clarity. It is not reflective of the content of the manuscript. The background did not provide the research gap to be filled. The study aim was not explicit enough (it was like a repetition of the study title). For methodology, pertinent elements like sample size, study instrument and how it was administered were missing. Reportage of the results should be improved. The conclusion is not in sync with the results

Authors response

We have snagged the title and provided the research gap in the background and the study aim has been expanded. We also added sample size, sample technique, study instrument to the study. The conclusion has been expanded. We have rewritten the reportage of the results. However, these modifications blew the abstract to more than the accepted 250 words format. See title and abstract 

Reviewer’s 2 comments

2. Introduction- The first sentence which reads- ‘Corona Virus infection (COVID 19) has now attained community infection and is now very common in families’- appears rather ambiguous. I think corona virus infection is not synonymous with COVID-19 (the latter is the disease caused by the former). Do the authors mean that the pandemic is now at the level of community transmission? Although the authors tried to highlight what is already known about the topic and the research question, the aim of the study as stated here (‘ascertaining the perception of mothers on the use of face mask in children in this COVID era’) is different from the aim as stated under the Abstract.

Authors response

We have deleted COVID 19 from the statement. We have replaced with pandemic.

We have corrected the aim in the abstract to be same with that in the introduction. See page 5 line 103-108

Reviewer’s comments

3. Methods- The authors revealed the study instrument here as structured self-administered questionnaires. A concise detail of the questionnaire items would have enlightened the reader. Would all the respondents have understood the items? An interviewer-administered questionnaire could have been a better approach.

Authors response

We used interviewer-administered questionnaire. We have added this in methodology. See page 7 line 148-150 

Reviewer’s comments

4. Results- The ‘practice’ aspect of this study namely ‘the wearing of face mask in children’ was based on proxy-report and not observational. Thus, the information provided by these mothers on the proportion of their children who did wear face mask may not be reliable. 

Authors response

The information given by the mother was supported with what was observed as the mothers were present with the children during the study. See page 6 line 130-131

Reviewer’s comments

This underscores the fact that an observational study of mother-child dyads would have been better. For instance, the authors reported that children aged 8 days to 1 year wore face masks. The practice is even contradictory to the reported WHO recommendation in children. 

Authors response

This was what was observed and reported. Another aspect of the study was providing information to the mothers on the correct use of facemasks. In fact, the finding of children aged 8 days to 1 year wearing face mask is an eye opener and will help alert the government to strengthen policy on acceptable age of wearing face mask and organize program geared on health education. See page 15 line 284-292 

Reviewer’s comments

The authors reported some socio-demographic variables as predictors of the practice of wearing face mask in children. Curiously, they were silent on the role of knowledge and perception. Maternal knowledge and perception ought to influence their practice of using face mask on their children

Authors response

This has been included. Thanks for the observation. How the perception of COVID-19 by the mothers was assessed has also been included. See table 2 and 3

Reviewer’s comments

5. Discussion- The content and flow of this section should be improved. The authors did not highlight any limitations of the current study. The conclusion was not precise enough but was too concise to have conveyed the message of the study.

Authors response

We have added limitation to the study, broadened the conclusion and we have mentioned recommendation for policies and program. See page17 line 329-342 

Once again, the authors thank you, the editorial team and the reviewers most sincerely for your critical review and suggestions which have improved the write up of this manuscript.

We hope this revised version of the manuscript will now be suitable for publication in your Journal.

Thank you Sir,

Dr. Chinawa Josephat

Corresponding Author

---

## [Decision Letter · Decision Letter 1]

15 Sep 2020

PONE-D-20-19085R1

Wearing of face mask by children and associated maternal factors; in the prevention of COVID 19: A multicentre study

PLOS ONE

Dear Dr. Chinawa,

Thank you for submitting your manuscript to PLOS ONE. After careful consideration, we feel that it has merit but does not fully meet PLOS ONE’s publication criteria as it currently stands. Therefore, we invite you to submit a revised version of the manuscript that addresses the points raised during the review process.

We look forward to receiving your revised manuscript.

Kind regards,

Francesco Di Gennaro

Academic Editor

PLOS ONE

Additional Editor Comments (if provided):

Dear authors,

follow reviewer suggestion you can improve your manuscript

Reviewers' comments:

Reviewer's Responses to Questions

**Comments to the Author**

1. If the authors have adequately addressed your comments raised in a previous round of review and you feel that this manuscript is now acceptable for publication, you may indicate that here to bypass the “Comments to the Author” section, enter your conflict of interest statement in the “Confidential to Editor” section, and submit your "Accept" recommendation.

Reviewer #1: (No Response)

Reviewer #2: (No Response)

2. Is the manuscript technically sound, and do the data support the conclusions?

Reviewer #1: Yes

Reviewer #2: Partly

3. Has the statistical analysis been performed appropriately and rigorously? 

Reviewer #1: (No Response)

Reviewer #2: No

4. Have the authors made all data underlying the findings in their manuscript fully available?

Reviewer #1: Yes

Reviewer #2: Yes

5. Is the manuscript presented in an intelligible fashion and written in standard English?

Reviewer #1: No

Reviewer #2: No

6. Review Comments to the Author

Reviewer #1: Thank you for addressing previously raised concerns. The authors have done a great job of addressing most of the issues. few suggestions are listed below.

1. There needs to be a thorough copy edit because there are several typographical errors. For instance, warding off is written as wading off

2. Please add the word Nigeria in the title.

Wearing of face mask by children and associated maternal factors; in the prevention of 2 COVID 19: A multicentre study in Nigeria

3. There is a lot of controversy on the use of face masks by children, especially in a situation where decisions bothering on who should wear the mask and at what age, still remain unknown. This statement does not convey the full guidance issues by WHO and Unicef of “doing no harm” and they are recommending that there is no need for mask for children < 5 years of age. However, some countries may contextualize it to 2 or 3 years. The authors need to highlight the correct wearing of masks at the outset

https://www.who.int/publications/i/item/WHO-2019-nCoV-IPC_Masks-Children-2020.1

Reviewer #2: (No Response)

7. PLOS authors have the option to publish the peer review history of their article (what does this mean?). If published, this will include your full peer review and any attached files.

Reviewer #1: No

Reviewer #2: No

---

## [Author Response · Author response to Decision Letter 1]

16 Sep 2020

REVIEWER #1: 

Thank you for addressing previously raised concerns. The authors have done a great job of addressing most of the issues. few suggestions are listed below.

1. There needs to be a thorough copy edit because there are several typographical errors. For instance, warding off is written as wading off

This has been corrected. See Page 13 line 239 

2. Please add the word Nigeria in the title.

Wearing of face mask by children and associated maternal factors; in the prevention of 2 COVID 19: A multicentre study in Nigeria

The second reviewer changed the title entirely but we have added the phrase “in Nigeria” to the title. See page 1 line 1-2

3. There is a lot of controversy on the use of face masks by children, especially in a situation where decisions bothering on who should wear the mask and at what age, still remain unknown. This statement does not convey the full guidance issues by WHO and Unicef of “doing no harm” and they are recommending that there is no need for mask for children < 5 years of age. However, some countries may contextualize it to 2 or 3 years. The authors need to highlight the correct wearing of masks at the outset

https://www.who.int/publications/i/item/WHO-2019-nCoV-IPC_Masks-Children-2020.1

The statement is now rewritten. See page 5 line 98-109

REVIEWER 2

REVIEWER’S COMMENTS ON REVISED MANUSCRIPT (PONE-D-20-19085R)

TITLE- Wearing of face mask by children and associated maternal factors; in the prevention of

COVID 19: A multicentre study 

General comments

Although the authors tried to address some of the issues pointed out in my previous review, the answers are not yet satisfactory. The revised manuscript still requires some modifications to improve its quality as there are obvious inconsistencies. The casuistry adopted by the authors in some of their responses has not been able to convince me on a number of specific issues which I have highlighted below

Specific issues and comments

1. Title: The current version of the title appears tenuous and does not reflect the message of the paper. Undoubtedly, the study was a KAP study on maternal perception of a preventive strategy for COVID-19 and not an observational study. I therefore suggest the authors modify the title thus- ‘Maternal perception of masking in children as a preventive strategy for COVID-19: a multicentre study’

We have changed the title to read “‘Maternal perception of masking in children as a preventive strategy for COVID-19: a multicentre study’”. The first reviewer wanted us to add……. in Nigeria. So the tile now reads “‘Maternal perception of masking in children as a preventive strategy for COVID-19 in Nigeria: a multicentre study” See page 1 line 1and 2

2. Abstract: The objective (or aim) as currently stated can be modified to read- ‘The study aimed to ascertain the perception of mothers on masking in children as a preventive strategy for COVID-19.’ 

This has been rewritten. Page 2 line 30-31

3. The summary of the Results here should be re-written for clarity. For instance, the statement- ‘A minor proportion of the children, 44.7% do wear face mask’- appears untenable in the context of this study. I presume the authors meant- ‘Minority (44.7%) of the mothers perceive masking in children as an appropriate measure for the prevention of COVID-19’. 

This has been corrected to read “Minority (44.7%) of the mothers perceive masking in children as an appropriate measure for the prevention of COVID-19’.” See page 2 line 36-39

The succeeding sentence- ‘The major reasons for the children not wearing face mask include perceived difficulty in breathing, 38.2% and that the child pulls it off, 29.3%.’- is suggested to read- ‘The major reasons given by the mothers for the inappropriateness of face mask in children include perceived difficulty in breathing (38.5%) and the child’s readiness to pull it off (29.3%).’

The statement now reads “The major reasons given by the mothers for the inappropriateness of face mask in children include perceived difficulty in breathing (38.5%) and the child’s readiness to pull it off (29.3%)” See page 2 line 36-39

 Again, one wonders how the authors arrived at the inferential statistics about the effect of maternal age, children’s age paternal educational status and maternal perception of COVID-19 on masking in children. The impression created here is that the authors actually observed masking in the children of these mothers. On the contrary, I suspect the authors merely evaluated the perception of selected mothers on the subject of masking in children as a preventive strategy for COVID-19. The conclusion that ‘The perception of mothers on the children wearing face mask is low’ should rather read-‘Maternal perception of masking in children as an appropriate strategy for preventing COVID-19 is adjudged low in this study. Right perception is significantly enhanced by maternal educational status etc.….’

We have now rewritten the conclusion to read “Maternal perception of masking in children as an appropriate strategy for preventing COVID-19 is adjudged low in this study. Right perception is significantly enhanced by maternal educational status” See page 3 line 53-55

4. Introduction: The opening sentence as corrected by the authors- ‘Corona Virus pandemic has now attained community infection and is now very common in families’- still appears ambiguous. What do they mean by the pandemic now attaining community infection? 

The statement “the pandemic now attaining community infection” has been expunged 

The statement after the study aim which reads- ‘It will also form the first data base on this topic in this region’- is considered a redundant statement and should be deleted.

This has been deleted

5. Materials and Methods: What do the authors mean by ‘The outcome variables in this study are the level of knowledge and perception of mothers on children wearing face mask’? (Lines 152-153). Nevertheless, this statement gives away the crux of the study as maternal knowledge/perception on masking in children as a preventive strategy for COVID-19. Thus, the authors should not be reporting about proportions of children who do wear face mask but rather on proportions of mothers with varying perceptions/knowledge on masking in children as a pandemic antidote. 

We have now rewritten the outcome variables to read “proportions of mothers with varying perceptions/knowledge on masking in children as a pandemic antidote” See page 8 line 164-167

 Again, this statement in lines 154 to 155 – ‘We used age, marital status, educational level, parity, husband’s educational level and occupation as independent variables’- is unclear. Do these variables-age, marital status and educational level-refer to the mothers? 

 This has been rewritten to read ‘We used mother’s and child’s age, marital status, educational level of both mother and father, parity, and occupation of father and mother as independent variables.” See page 8 line 15-167

 Results: The authors in trying to justify the fact that they observed the children of the mothers wearing face mask stated ‘The information given by the mother was supported with what was observed as the mothers were present with the children during the study. See page 6 line 130-131’. Did the authors study mother-child dyads? It was not stated in this study. What form of face masks were worn by the children that were neonates and infants? Do the authors have customized face masks for such age groups in their region? The WHO dissuades the use of face masks in children less than 2 years. Recently, the age bar has been raised to 12 years. 

 We did not focus on mother child dyad of wearing of face mask, though mothers were observed in the study, masked. However, the National COVID-19 protocol made it mandatory for adults to wear facemasks in public space, but this was strictly enforced by health facilities and financial institutions. Our observation of child wearing face mask necessitated this study. However, there was no such clear rule for children. See page 7 line 142-150. The home-made face mask was used by neonates and infants’ children in this study. 

6. Discussion: The revised discussion is not yet robust enough. Several syntax errors and redundant sentences precluded its clarity. For instance, the opening sentence-‘This study has gone a long way to show the importance of prevention of COVID 19 in children by wearing of face mask’ is ambiguous.

We have rewritten the statement to now read “This study has shown varying maternal perceptions/knowledge on masking in children as a pandemic antidote.” See page 13 line 232.We have added some items in discussion. See Discussion section

 Several of such statements are littered through this section. 

We have tried our best to correct all syntax errors

Even the study limitation is unrealistic. How could the authors not have observed if the mothers wore face masks since they claimed they observed masking in their children? If it was an oversight, it further confirms my belief that the study was fundamentally conducted on evaluating maternal perception/knowledge on masking in children.

We have deleted the study limitation since the study was fundamentally conducted on evaluating maternal perception/knowledge on masking in children. See page 17 line 350-351

---

## [Decision Letter · Decision Letter 2]

29 Sep 2020

PONE-D-20-19085R2

Maternal perception of masking in children as a preventive strategy for COVID-19 in Nigeria: a multicentre study.

PLOS ONE

Dear Dr. Chinawa,

Thank you for submitting your manuscript to PLOS ONE. After careful consideration, we feel that it has merit but does not fully meet PLOS ONE’s publication criteria as it currently stands. Therefore, we invite you to submit a revised version of the manuscript that addresses the points raised during the review process.

Please submit your revised manuscript by 20 October If you will need more time than this to complete your revisions, please reply to this message or contact the journal office at plosone@plos.org. Please include the following items when submitting your revised manuscript:

We look forward to receiving your revised manuscript.

Kind regards,

Francesco Di Gennaro

Academic Editor

PLOS ONE

Additional Editor Comments (if provided):

Dear authors follow minor comments of reviewers to improve your article

Reviewers' comments:

Reviewer's Responses to Questions

**Comments to the Author**

1. If the authors have adequately addressed your comments raised in a previous round of review and you feel that this manuscript is now acceptable for publication, you may indicate that here to bypass the “Comments to the Author” section, enter your conflict of interest statement in the “Confidential to Editor” section, and submit your "Accept" recommendation.

Reviewer #1: (No Response)

Reviewer #2: All comments have been addressed

2. Is the manuscript technically sound, and do the data support the conclusions?

Reviewer #1: Yes

Reviewer #2: Partly

3. Has the statistical analysis been performed appropriately and rigorously? 

Reviewer #1: Yes

Reviewer #2: Yes

4. Have the authors made all data underlying the findings in their manuscript fully available?

Reviewer #1: Yes

Reviewer #2: Yes

5. Is the manuscript presented in an intelligible fashion and written in standard English?

Reviewer #1: Yes

Reviewer #2: No

6. Review Comments to the Author

Reviewer #1: A minor proportion of the children, 44.7% wore face 194 masks which were all home-made.

According to Table 1 this should be 43.5% 44.7% are females

230. Children less than 2 years may not 231 benefit from wearing face mask but may rather be in danger. –Please clarify this statement. I am assuming that you are suggesting that wearing a mask maybe more high-risk for them

Reviewer #2: (No Response)

7. PLOS authors have the option to publish the peer review history of their article (what does this mean?). If published, this will include your full peer review and any attached files.

Reviewer #1: No

Reviewer #2: No

---

## [Author Response · Author response to Decision Letter 2]

29 Sep 2020

Re- Maternal perception of masking in children as a preventive strategy for COVID-19 in

Nigeria: a multicentre study.

REVIEWER’S COMMENTS

 Review Comments to the Author

Reviewer #1: A minor proportion of the children, 44.7% wore face 194 masks which were all home-made.

According to Table 1 this should be 43.5% 44.7% are females

This is now corrected. See Page 9,line 197-198

230. Children less than 2 years may not 231 benefit from wearing face mask but may rather be in danger. –Please clarify this statement. I am assuming that you are suggesting that wearing a mask maybe more high-risk for them

We have modified the statement. See Page 13 line 238

The authors have done pretty well in addressing the concerns raised in the previous review. However, there are still some specific issues that need their attention to further improve this manuscript

1. Under the ABSTRACT, some statements need correction. I suggest these lines should read- ‘The frequent reasons given by majority (55.3%) of the mothers for the inappropriateness of face mask in children…….’ ‘A significantly higher proportion (64.2%) of the children…..would wear face masks when compared with 31.7% of those whose mothers were < 30 years of age (x2 =28.632, p<0.001).’ Similarly, a significantly higher proportion (51.0%) of the children who were more than one year of age would wear face mask when compared with 20.5% of those aged eight days to one year (x2=19.441, p<0.001).’

This has been corrected. See Page 2 line 37-45

2. Lines 118-119 (Under INTRODUCTION), the study aim should be written as it appears in the ABSTRACT.

We have written the study aim as it appeared in the abstract. See Page 6 line 119-120

3. Under RESULTS, lines 191-194 should reflect what is in the Abstract. As currently written, the message is inconsistent with that in the Abstract.

We have reflected statements in results to be in keeping with that in abstract.

See page 9 line 193-198

 Under DISCUSSION, lines 230-231 read ‘Children less than 2 years may not benefit….may rather be in danger.’ The authors should add ‘….may rather be in danger of suffocation’.

We have added the phrase may rather be at risk of suffocation’. See Page 13 line 238

4. Line 252 should be re-written thus- ‘Few mothers in this study…’ 

This has been added. See Page 14 line 260

5. Lines 274-276- In line with my previous suggestions, ‘wear face masks’ should rather read ‘would wear face masks’ to be in keeping with the study title of maternal perception of masking in children. 

This has been effected. See Page 15 Line 282-283

6. Similarly, lines 295-298 should be corrected accordingly.

This has been corrected. See Page 15 Line 299

7. For line 313- delete this redundant statement-‘These assertion is also upheld in our study.’

This has been deleted

8. The study limitation as currently stated is ambiguous. The authors should rather state that they did not evaluate the concordance of maternal perception of masking with actual wearing of face mask in their children. 

We have added this in limitation. See Page 18 line 350-351

---

## [Decision Letter · Decision Letter 3]

13 Oct 2020

PONE-D-20-19085R3

Maternal perception of masking in children as a preventive strategy for COVID-19 in Nigeria: a multicentre study.

PLOS ONE

Dear Dr. Chinawa,

Thank you for submitting your manuscript to PLOS ONE. After careful consideration, we feel that it has merit but does not fully meet PLOS ONE’s publication criteria as it currently stands. Therefore, we invite you to submit a revised version of the manuscript that addresses the points raised during the review process.

We look forward to receiving your revised manuscript.

Kind regards,

Francesco Di Gennaro

Academic Editor

PLOS ONE

Additional Editor Comments (if provided):

Dear Authors follow reviewer suggest (minor suggestions) to improve your manuscript

Reviewers' comments:

Reviewer's Responses to Questions

**Comments to the Author**

1. If the authors have adequately addressed your comments raised in a previous round of review and you feel that this manuscript is now acceptable for publication, you may indicate that here to bypass the “Comments to the Author” section, enter your conflict of interest statement in the “Confidential to Editor” section, and submit your "Accept" recommendation.

Reviewer #1: (No Response)

Reviewer #2: (No Response)

2. Is the manuscript technically sound, and do the data support the conclusions?

Reviewer #1: (No Response)

Reviewer #2: Partly

3. Has the statistical analysis been performed appropriately and rigorously? 

Reviewer #1: (No Response)

Reviewer #2: Yes

4. Have the authors made all data underlying the findings in their manuscript fully available?

Reviewer #1: (No Response)

Reviewer #2: Yes

5. Is the manuscript presented in an intelligible fashion and written in standard English?

Reviewer #1: (No Response)

Reviewer #2: No

6. Review Comments to the Author

Reviewer #1: Thank you for addressing the comments. Please clarify the following

Table 1 shows the wearing of face mask by the children. Minority (44.7%) of the mothers

193 perceived masking in children as an appropriate measure for the prevention of COVID-19. The

194 frequent reasons given by majority (55.3%) of the mothers for the inappropriateness of face

195 mask in children included perceived difficulty in breathing (38.5%) and the child’s readiness

196 to pull it off (29.3%)

In table1- 44.7 and 55.3 refers to gender - male and female. Please clarify if it is for mothers? the table is titles Child wearing a mask so its very confusing.

Reviewer #2: General comments:

Although the authors have progressively improved the quality of this manuscript by their revisions, I would urge them to be more painstaking in responding to the specific issues of concern previously raised. This will not only eliminate inconsistencies within the manuscript but also improve the lexicon of their written English.

Specific comments:

1. Abstract- In line 41, the repeated figure (64.2%) should be deleted.

2. In my previous reviews, I pointed out that the manuscript has grammar and syntax errors. I suggested an English editing to be done. However, it appears the authors have disregarded this suggestion. This is evident in some errors noted as follows: ‘was’ used instead of ‘were’ (Line 77), ‘is’ used instead of ‘are’ (Line 78), ‘In China’ instead of ‘in China’ (Line 90) and ‘intensified program’ used instead of ‘intensified programs’, etc. Furthermore, I guess the statement in line 145 should read ‘we did not focus on mother-child dyads while assessing maternal perception and the practice of wearing face masks, though…..’ In fact, this assertion throws up an inconsistency with lines 178-179 where the authors stated that a ‘Yes’ answer on wearing face mask was confirmed by their observation of masking in children during data collection. How could the authors not have focused on mother-child pairs in the study when they reported masking in both the mothers and their children? This again negates the study limitation in lines 350-351 which states ‘We did not evaluate the concordance of maternal perception of masking with actual wearing of face mask in their children.’ Let the authors correct this inconsistency

3. Finally, I urge the authors to subject the manuscript to English editing. I suggest they could use the ‘grammarly’ software if they cannot utilize the available English editing services

7. PLOS authors have the option to publish the peer review history of their article (what does this mean?). If published, this will include your full peer review and any attached files.

Reviewer #1: No

Reviewer #2: No

---

## [Author Response · Author response to Decision Letter 3]

15 Oct 2020

Reviewer #1: Thank you for addressing the comments. Please clarify the following

Table 1 shows the wearing of face mask by the children. Minority (44.7%) of the mothers 193 perceived masking in children as an appropriate measure for the prevention of COVID-19. The194 frequent reasons given by majority (55.3%) of the mothers for the inappropriateness of face

195 mask in children included perceived difficulty in breathing (38.5%) and the child’s readiness 196 to pull it off (29.3%)

This statement has been expunged and the section rewritten. See page 9 line 192-195 

In table1- 44.7 and 55.3 refers to gender - male and female. Please clarify if it is for mothers? the table is titles Child wearing a mask so its very confusing.

This has been corrected. See page 9 line 192. See also Table one

Reviewer #2: General comments:

Although the authors have progressively improved the quality of this manuscript by their revisions, I would urge them to be more painstaking in responding to the specific issues of concern previously raised. This will not only eliminate inconsistencies within the manuscript but also improve the lexicon of their written English.

Specific comments:

1. Abstract- In line 41, the repeated figure (64.2%) should be deleted.

This has been deleted. 

2. In my previous reviews, I pointed out that the manuscript has grammar and syntax errors. I suggested an English editing to be done. However, it appears the authors have disregarded this suggestion. This is evident in some errors noted as follows: ‘was’ used instead of ‘were’ (Line 77), ‘is’ used instead of ‘are’ (Line 78), ‘In China’ instead of ‘in China’ (Line 90) and ‘intensified program’ used instead of ‘intensified programs’, etc. 

We have sent the paper to Grammarly for English editing. We believe that the paper looks better than the previous one. The corrected syntax is as highlighted. See page 4 and 5

Furthermore, I guess the statement in line 145 should read ‘we did not focus on mother-child dyads while assessing maternal perception and the practice of wearing face masks, though.’ 

This has been corrected. See page 7 line 146-148

In fact, this assertion throws up an inconsistency with lines 178-179 where the authors stated that a ‘Yes’ answer on wearing face mask was confirmed by their observation of masking in children during data collection. How could the authors not have focused on mother-child pairs in the study when they reported masking in both the mothers and their children

Though we reported masking in both mothers and their children, but unfortunately, we did not ascertain the practice of masking on mother-child pair as this was not factored in the questionnaire. We are sorry for this.

This again negates the study limitation in lines 350-351 which states ‘We did not evaluate the concordance of maternal perception of masking with actual wearing of face mask in their children.

The study limitation now reads “We did not ascertain actual practice of masking on mother-child pair”. See page 17 line 343

’ Let the authors correct this inconsistency

We have tried to correct these inconsistencies

3. Finally, I urge the authors to subject the manuscript to English editing. I suggest they could use the ‘grammarly’ software if they cannot utilize the available English editing services

We have subjected the article to English editing using Grammarly.

---

## [Decision Letter · Decision Letter 4]

9 Nov 2020

Maternal perception of masking in children as a preventive strategy for COVID-19 in Nigeria: a multicentre study.

PONE-D-20-19085R4

Dear Dr. Chinawa,

We’re pleased to inform you that your manuscript has been judged scientifically suitable for publication and will be formally accepted for publication once it meets all outstanding technical requirements.

Kind regards,

Francesco Di Gennaro

Academic Editor

PLOS ONE

Additional Editor Comments (optional):

Dear Authors congratulations

Reviewers' comments:

Reviewer's Responses to Questions

**Comments to the Author**

1. If the authors have adequately addressed your comments raised in a previous round of review and you feel that this manuscript is now acceptable for publication, you may indicate that here to bypass the “Comments to the Author” section, enter your conflict of interest statement in the “Confidential to Editor” section, and submit your "Accept" recommendation.

Reviewer #1: All comments have been addressed

Reviewer #2: All comments have been addressed

2. Is the manuscript technically sound, and do the data support the conclusions?

Reviewer #1: Yes

Reviewer #2: Yes

3. Has the statistical analysis been performed appropriately and rigorously? 

Reviewer #1: Yes

Reviewer #2: Yes

4. Have the authors made all data underlying the findings in their manuscript fully available?

Reviewer #1: Yes

Reviewer #2: Yes

5. Is the manuscript presented in an intelligible fashion and written in standard English?

Reviewer #1: Yes

Reviewer #2: (No Response)

6. Review Comments to the Author

Reviewer #1: Please correct the following sentence.

Do your child wear a face mask

It should read - Does your child wear a mask

Reviewer #2: Thank you for addressing the concerns I raised about your manuscript. During the galley proof, you can further improve the lexicon

7. PLOS authors have the option to publish the peer review history of their article (what does this mean?). If published, this will include your full peer review and any attached files.

Reviewer #1: No

Reviewer #2: **Yes: **Samuel Uwaezuoke

---

## [Editor Report · Acceptance letter]

12 Nov 2020

PONE-D-20-19085R4 

Maternal perception of masking in children as a preventive strategy for COVID-19 in Nigeria: a multicentre study. 

Dear Dr. Chinawa:

I'm pleased to inform you that your manuscript has been deemed suitable for publication in PLOS ONE. Congratulations! Your manuscript is now with our production department. 

Kind regards, 

on behalf of

Dr. Francesco Di Gennaro 

Academic Editor

PLOS ONE